# Research on the Design Strategy of Healing Products for Anxious Users during COVID-19

**DOI:** 10.3390/ijerph19106046

**Published:** 2022-05-16

**Authors:** Fan Wu, Yang-Cheng Lin, Peng Lu

**Affiliations:** 1Department of Industrial Design, National Cheng Kung University, Tainan 70101, Taiwan; wuxiaofan999@gmail.com; 2Department of Industrial Design, Xi’an Academy of Fine Arts, Xi’an 710000, China

**Keywords:** healing product, anxious user, design strategy, evaluation criteria, design guidelines

## Abstract

With the spread of COVID-19 worldwide, various travel restrictions are becoming a significant trigger for anxiety. Although healing products could relieve anxiety, few scholars have proposed a practical design strategy. Therefore, we offer a design strategy for healing products that includes three phases: preparation, analysis, and verification. In Phase 1, 20 people with moderate or high anxiety are invited to rate 100 samples. Then, FCM is used to obtain representative samples. In Phase 2, a three-layer diagram (incl. the upper, middle, and lower layers) of healing products is obtained using the evaluation grid method. Subsequently, the middle layer is considered evaluation criteria. Additionally, 18 items in the lower layer are considered design guidelines. In Phase 3, we invite two teams to develop innovative designs based on design guidelines and personal experience, generating four alternatives. Finally, four alternatives and four healing commodities are evaluated using grey relation analysis and perceptual questionnaires. The consistency of both evaluations could confirm the validity of the evaluation criteria. The alternatives generated based on the design guidelines are better than other alternatives, demonstrating the effectiveness of the design guidelines. The design strategy is beneficial for developing and evaluating healing products to alleviate people’s anxiety during COVID-19.

## 1. Introduction

Since 2019, COVID-19 has swept the world [1]. The government affected by the epidemic urged residents not to go out as much as possible to prevent the virus from spreading among people [2]. During this period, the masses actively responded to the government’s request to reduce travel via the home office, online teaching, and centralized procurement in the community. However, long-term living in a limited space can cause people to feel anxious [3]. Qiu et al. [4] and Trapp et al. [5] found that people are more prone to anxiety during travel restrictions. Specifically, women are more prone to anxiety than men, and people aged 18 to 30 and over 60 are more prone to anxiety. This part of the population needs to consider the children’s physical and mental health, whether they can finish their studies on time, whether their income is stable, whether their medical treatment is convenient, etc., therefore they would have a sense of anxiety. In addition, Bao et al. [6] and Cha et al. [7] showed that as more and more countries suffer from the repeated invasion of COVID-19 and its variants, people are increasingly worried that they or their family members will be infected, which leads to anxiety. Currently, the epidemic covers all countries and people’s anxiety has increased. If anxiety accumulates to a certain level, it will directly affect people’s physical and mental health. Therefore, it is necessary to alleviate people’s anxiety through effective means.

Many scholars focused on how to relieve people’s anxiety and negative emotions. Amber [8] proposed that harmonious color combinations could drive away people’s anxiety. Bradt et al. [9] believed that the melody of music could provide comfort and hope to people who are depressed, thereby alleviating anxiety. Peng and Wu [10] pointed out that people would have a relaxed mental state when interacting with objects of certain materials, which will eliminate anxiety. Hepworth et al. [11] showed that people could alleviate negative emotions by eating sweets. Based on the above analysis, we found that people could alleviate negative emotions and anxiety after receiving visual, auditory, tactile, and taste stimuli. In addition, Kotler [12] pointed out that although smell is the most obscure of the five senses, it could still affect people’s behavior and psychological feelings. Therefore, if we want to alleviate people’s anxiety through external intervention, we must start from the five senses of human beings.

Healing refers to the process of relaxing people’s moods or releasing stress through effective means [13]. Chen [14] pointed out that people can vent their accumulated anxiety and negative emotions through unique healing methods. Chen [15] believes that the essence of healing is to help people transfer negative emotions and make people feel comfortable and happy. Currently, the healing industry has developed many branches, including healing literature, healing music, healing fluid art, and healing products. Among them, healing products are regarded as the most popular healing method by multi-sensory stimulation. In the visual-led healing research, Syu [16] divided the shapes of healing products into animals, plants, figures, and non-biology and pointed out that animals are the primary source of reference for healing products. Amber [8] believed that the color matching of healing products should use warm and soft colors instead of bright colors. In the tactile-led healing research, Peng and Wu [10] pointed out that when people interact with healing products, they could increase hand flexibility and exercise muscles, thereby alleviating anxiety and tension. In the auditory-led healing research, Bradt et al. [9] believed that some healing products that produce musical melodies could divert the attention of anxious patients and help them achieve inner peace. Furthermore, in addition to a few healing products that can emit odor and be eaten, taste and smell are mostly metaphorical in healing products. For example, the packaging designed by Japanese designer Naoto Fukasawa for fruit juice can remind people of the taste and smell of fruit.

In the context of the rapid spread of COVID-19, more and more people feel anxious. As mentioned above, although many scholars analyze existing healing products based on the five senses of human beings, few scholars have proposed a specific design strategy for healing products. Therefore, this research adopts effective research methods to provide detailed design guidelines and evaluation criteria for healing products to alleviate people’s anxiety. Generally, the target groups of existing healing products are mainly special groups, such as children [17], older people [18], and single women [19]. However, this research takes the anxious group in the context of COVID-19 as the target group.

In conclusion, those healing products are considered an effective means to relieve anxiety. Thus, this study integrates scientific and appropriate research methods to propose a practical design strategy for healing products. Precisely, the design strategy consists of three phases, namely preparation, analysis, and verification. Finally, we could obtain a set of evaluation criteria and design guidelines for healing products. This paper is organized as follows: Section 2 describes the research methods involved in this research. Section 3 explains the research process and the specific implementation steps. Section 4 elaborates on the construction process of the design strategy. Finally, the last section provides the discussions and conclusions.

## 2. Theoretical Background

This section aims to clarify the literature and lay the foundation for the next steps. The detailed theory and research methods are described below.

### 2.1. State-Trait Anxiety Inventory

Anxiety is an irritable emotion caused by excessive worry about one’s own or relatives’ safety, prospects, and destiny. Anxious people are prone to fear, panic, tension, nervousness, and other psychological states. Generally, anxiety is an emotional state divided into short-term anxiety and long-term anxiety. To measure people’s anxiety levels, scholars in psychology have proposed many measures, including the depression anxiety stress scale (DASS) [20], the self-rating depression scale (SDS) [21], the self-rating anxiety scale (SAS) [22], and the state-trait anxiety inventory (STAI) [23]. Among them, STAI is the most commonly used method for measuring anxiety in applied psychology. STAI consists of two parts, each with 20 questions [24]. The first part is used to assess how one feels at the moment and the second part is used to evaluate the state of how one generally feels. A person is classified as anxious when the overall test score exceeds 80 [25]. In this paper, we use STAI to screen out people with moderate or high anxiety as testees in the preparation phase of the design strategy.

### 2.2. Fuzzy C-Means (FCM) Clustering

Cluster analysis belongs to unsupervised machine learning and is often used to cluster similar information in data [26]. Dunn [27] first proposed the Hard C-means clustering method. Subsequently, Bezdek (1981) proposed Fuzzy C-means (FCM) based on the Hard C-means clustering method. FCM is an algorithm that uses fuzzy theory to solve the optimal cluster problem; that is, the membership of each sample point to all cluster centers is obtained by optimizing the objective function to achieve the purpose of automatic classification of samples. Compared with other clustering methods, FCM has the characteristics of fast convergence speed and objective clustering results. After years of development, FCM has been widely used in various fields, such as computing [28], management [29], and medicine [30]. In this paper, we use FCM to classify the collected research samples during the preparation phase of the design strategy.

The minimizing objective function of FCM [31] is shown in Equation (1).
(1)$$J_{m}=\sum_{i=1}^{N}\sum_{j=1}^{c}u_{ij}^{m}{\left|\left|x_{i}-c_{j}\right|\right|}^{2},\;1\leq m\leq \infty$$
where *U* = [$u_{ij}$] is the fuzzy membership matrix, $u_{ij}$ is the membership of the sample *j* to the category *i*, *m* is the fuzzy constant (m≥1), $x_{i}$ is the data of dimension *i* of data *x*, $c_{j}$ represents cluster center, and ||*|| indicates the similarity between the sample data and the cluster center. In addition, for any sample, the sum of membership degrees is 1, as shown in Equation (2).
(2)$$\sum_{j=1}^{C}u_{ij}=u_{1,1}+u_{2,1}\ldots =1$$

In the clustering process, the membership degrees of samples belonging to each cluster are different. In other words, the closer a sample is to the cluster center point the higher its membership and the farther it is the lower its membership [32]. FCM is a process of repeatedly calculating the minimizing objective function and continuously updating the membership degree ($u_{ij}$) and cluster center ($c_{j}$), as shown in Equations (3) and (4).
(3)$$u_{ij}=\frac{1}{\sum_{k=1}^{c}{\left(\frac{\left|\left|x_{i}-c_{j}\right|\right|}{\left|\left|x_{i}-c_{j}\right|\right|}\right)}^{\left(\frac{2}{m-1}\right)}}$$
(4)$$c_{j}=\frac{\sum_{i=1}^{N}u_{ij}^{m}*x_{i}}{\sum_{i=1}^{N}u_{ij}^{m}}$$

In addition, the condition for the repeated calculation to stop is when $max_{ij}${||$u_{ij}^{\left(k+1\right)}-$|$u_{ij}^{\left(k\right)}$||}<ε, and the termination condition where ε is between 0 and 1, *k* as iterative step, this process will converge to a local minimum of the objective function $J_{m}$.

### 2.3. Miryoku Engineering and Evaluation Grid Method (EGM)

Miryoku Engineering was proposed by Japanese scholar Masato Ujigawa, aiming to design attractive products based on consumer preferences [33]. Attractive refers to the attractive positive factors of a product, which carry users’ needs at all levels of the product. Generally, the attractive factor of a product could be obtained through the evaluation grid method (EGM). EGM is an effective research method of Miryoku Engineering developed by Japanese scholars Junichiro Sanuiand based on the psychology repertory grid method [34]. Specifically, a three-layer diagram of product attractiveness factors could be obtained through EGM. The execution steps of EGM are as follows [35]: First, collect product images or buy the products as stimuli. Second, use stimuli to conduct in-depth interviews with subjects to obtain an individual’s evaluation information. Finally, organize the evaluation information of all subjects to obtain a three-layer diagram, including abstract evaluation items (upper layer), original evaluation items (middle layer), and concrete evaluation items (lower layer). In real-life, EGM has been widely used in design problems such as App Icons [36], online teaching [37], and product form [38]. In this paper, to propose a practical healing product design strategy, we need to use EGM to analyze the attractiveness factors of existing healing products during the analysis phase of the design strategy.

### 2.4. Quantification Theory Type I (QTTI)

Quantification theory Type I (QTTI) uses multiple regression analysis to establish the mapping relationship between the independent variable *x* and the dependent variable *y* to realize the prediction of the dependent variable *y* [39]. In previous studies, many scholars used QTTI and questionnaires to analyze the three-layer diagram generated by EGM and then clarified the importance and credibility of each attractive factor according to the category score and partial correlation coefficient in the analysis results [35,36,37,38]. In this paper, since the lower layer in the three-layer diagram is the most concrete attractive factor, we use QTTI to obtain the items with higher category scores in the lower layer as the design guidelines.

### 2.5. Analytical Hierarchy Process (AHP)

The analytical hierarchy process (AHP) was developed by Thomas Saaty [40] to help decision makers choose the best solution under the uncertain environment of multi-criteria. AHP uses pairwise comparison to obtain the relative weights of multiple alternatives under multiple evaluation criteria. The specific implementation steps [35] are as follows.
Define the decision problem: determine the purpose of this decision problem and list the evaluation items to be analyzed;Build a hierarchy: the top layer of the hierarchy is the purpose of this decision problem and the bottom layer is the evaluation items;Create a pairwise comparison matrix: each evaluation item is compared to other items based on a relative scale table (see Table 1);Get the relative weights of all evaluation items: the relative weight of each evaluation item is obtained using the geometric mean method;Consistency testing: the calculation results are checked according to the consistency relation (*C.R.*), as shown in Equation (5).
(5)$$C.R.=\frac{C.I.}{R.I.},\;\;C.I.=\frac{\lambda_{max}-n}{n-1}$$ where *C.I.* is the consistency index, *R.I.* is the random index (see Table 2), *n* is the order of the matrix, and $\lambda_{max}$ is the largest eigenvalue of the matrix. When *C.R.* < 0.1, it means that the judgment matrix is acceptable.

**Table 1 ijerph-19-06046-t001:** Evaluation measurement and relative definition of the analytic hierarchy process.

Evaluation Measurement	Definition
1	Equal importance
3	Slight importance
5	Essential importance
7	Very strong importance
9	Absolute importance
2, 4, 6, 8	Intermediate values

**Table 2 ijerph-19-06046-t002:** Table of random indexes.

Order N	3	4	5	6	7	8	9	10	11
*R.I.*	0.58	0.9	1.12	1.24	1.32	1.44	1.45	1.49	1.51

AHP is an objective and practical research method [41]. Currently, AHP is widely used in decision-making problems in various disciplines such as product design [42,43,44], architecture [45], materials science [46], oceanography [47]. The abstraction level of the middle layer in the three-layer diagram is between the upper layer and the lower layer. Thus, this study takes the attractiveness factor in the middle layer as the evaluation criteria of healing products and then uses AHP to obtain the relative weight of each evaluation criterion.

### 2.6. Grey Relational Analysis (GRA)

Grey system theory was proposed by Deng [48] and could be used to solve problems such as system analysis, modeling, prediction, decision making, control, and evaluation. Grey relational analysis (GRA) is a part of grey systems and is suitable for dealing with uncertain associations between things in a system or between factors in a system to the main behavior [49]. The execution steps and algorithms of GRA are described below [50].

Step 1: Determine the ideal optimal target sequence.

The raw data about things in a system can be represented as a set of comparative sequences after initializing, as shown in Equation (6). According to the principle of relative optimization, an ideal optimal target sequence is extracted from the comparison sequence as the reference sequence $x_{0}$, as shown in Equation (7).
(6)$$\begin{array}{cc} \begin{array}{c} x_{1}=\left\{x_{1}\left(1\right),x_{1}\left(2\right),\cdots ,x_{1}\left(m\right)\right\}\\ x_{2}=\left\{x_{2}\left(1\right),x_{2}\left(2\right),\cdots ,x_{2}\left(m\right)\right\}\\ \vdots \\ x_{n}=\left\{x_{n}\left(1\right),x_{n}\left(2\right),\cdots ,x_{n}\left(m\right)\right\} \end{array} & ,\;m\geq 3;n\geq 3 \end{array}$$
(7)$$x_{0}=\left\{x_{0}\left(1\right),x_{0}\left(2\right),\cdots ,x_{0}\left(m\right)\right\}$$
where the values of $x_{0}\left(1\right)$ to $x_{0}\left(m\right)$ are as shown in Equation (8).
(8)$$\begin{array}{cc} \begin{array}{c} x_{0}\left(1\right)=max\;x_{i}\left(1\right)\\ x_{0}\left(2\right)=max\;x_{i}\left(2\right)\\ \vdots \\ x_{0}\left(m\right)=max\;x_{i}\left(m\right) \end{array} & ,\;i=1,2,\cdots ,n \end{array}$$

Step 2: Calculate the grey relational coefficient.

The grey relational coefficient can judge the correlation between the comparison sequences $x_{i}$ and the reference sequence $x_{0}$. First, the sequence of differences $\Delta_{i}\left(k\right)$ between each comparison sequence and the reference sequence is calculated separately, as shown in Equation (9).
(9)$$\Delta_{i}\left(k\right)=\left|x_{0}\left(k\right)-x_{i}\left(k\right)\right|,\;k\in \left\{1,2,\cdots ,m\right\}$$

Subsequently, the values of $\Delta_{max}$ and $\Delta_{min}$ are calculated in Equation (10).
(10)$$\begin{array}{c} \Delta_{max}=\underset{i\in I}{\max}\underset{k}{\;\max}\left|x_{0}\left(k\right)-x_{i}\left(k\right)\right|\\ \Delta_{min}=\underset{i\in I}{\min}\underset{k}{\;\min}\left|x_{0}\left(k\right)-x_{i}\left(k\right)\right| \end{array}$$

Finally, the grey relational coefficient $\xi_{i}\left(k\right)$ can be obtained by substituting $\Delta_{i}\left(k\right)$, $\Delta_{max}$, $\Delta_{min}$ into Equation (11).
(11)$$\xi_{i}\left(k\right)=\frac{\underset{i\in I}{\min}\underset{k}{\;\min}\left|x_{0}\left(k\right)-x_{i}\left(k\right)\right|+\rho \underset{i\in I}{\max}\underset{k}{\;\max}\left|x_{0}\left(k\right)-x_{i}\left(k\right)\right|}{\left|x_{0}\left(k\right)-x_{i}\left(k\right)\right|+\rho \underset{i\in I}{\max}\underset{k}{\;\max}\left|x_{0}\left(k\right)-x_{i}\left(k\right)\right|}$$
where ρ is the discrimination coefficient and the value range is 0≤ρ≤1. Generally, ρ is set to 0.5 in practical applications.

Step 3: Calculate the grey relation.

Based on the grey relational coefficient, the grey relation between the comparison sequences $x_{i}=\left\{x_{1},x_{2},\cdots ,x_{n}\right\}$ and the reference sequence $x_{0}$ can be further calculated. The average value of the grey relational coefficient $\xi_{i}\left(k\right)$ is taken as the value of the grey relation $r_{i}$, as shown in Equation (12).
(12)$$r_{i}=\frac{1}{n}\sum_{k-1}^{m}\xi_{i}\left(k\right),\;i=1,2,\cdots ,n$$

Step 4: Obtain the priority of comparative sequences.

All comparison sequences are sorted according to the grey relation obtained in Step 3. The larger the grey relation value between a comparison sequence and the reference sequence the closer the comparison sequence is to the ideal optimal target sequence.

GRA is widely used to evaluate schemes [51,52]. In this paper, in the verification phase of the design strategy we use the weighted items from the middle layer as the evaluation criteria and then use GRA to evaluate the healing commodity and the newly designed schemes. Finally, we compare the evaluation results with the results of the perceptual evaluation.

## 3. Implementation Procedures

The design strategy for healing products proposed in this study consists of three phases: preparation, analysis, and verification. Based on the research method described in Section 2, the specific implementation steps are as follows and the research process is shown in Figure 1.

Preparation phase:

Step 1: people who are in quarantine or have experienced quarantine during COVID-19 are invited to fill in the STAI questionnaire, thereby screening out people with moderate or high anxiety as testees.

Step 2: collect 100 healing commodities as research samples through expert recommendation and internet research and construct an information table including sample images, functions, and materials.

Step 3: a set of essential characteristics suitable for describing the healing contribution of samples is proposed based on the five senses (i.e., sight, hearing, touch, smell, taste).

Step 4: invite testees to score 100 samples using essential characteristics on a scale of 0 to 1.

Step 5: based on the scoring data, all samples are classified using FCM and the three schemes closest to the center point in each cluster are taken as the representative samples of this cluster.

Step 6: purchase all representative samples through online sales platforms.

Analysis phase:

Step 1: invite half of the testees to experience the purchased representative samples and then conduct in-depth interviews with the testees using the evaluation grid method.

Step 2: collate the interview information to obtain a three-layer diagram about healing products, namely the abstract evaluation items (upper layer), the original evaluation items (middle layer), and the concrete evaluation items (lower layer).

Step 3: use the items in the middle layer as evaluation criteria for healing products and then use AHP to obtain the relative weight of each evaluation criterion.

Step 4: set up a questionnaire using the three-layer diagram and invite all testees to complete the questionnaire.

Step 5: use QTTI to analyze the questionnaire results and use the items with higher category scores in the lower layer as design guidelines.

Verification phase:

Step 1: invite two groups of industrial design students to carry out innovative designs based on design guidelines and personal experience, and the newly designed schemes are regarded as the innovation group.

Step 2: the one scheme in each cluster closest to the center point is regarded as the reference group.

Step 3: invite half of the testees to use the evaluation criteria to score the schemes in the innovation group and the reference group and then use the GRA to obtain the priority of all schemes.

Step 4: invite the other half of the testees to use the perceptual evaluation questionnaire to score the schemes in the innovation group and the reference group and then obtain the priority of all schemes.

Step 5: verify whether the two evaluation results are consistent.

Step 6: verify whether the schemes generated according to the design guidelines are better than those generated according to personal experience.

## 4. Empirical Study

Based on the implementation steps described in Section 3, this section presents the detailed construction process of the healing product design strategy for anxious users during COVID-19.

### 4.1. Phase 1: Preparation

#### 4.1.1. Testees

The continued spread of COVID-19 has led to the normalization of centralized isolation or home isolation. During the quarantine period, the closed environment could make people feel anxious. Therefore, this study invited people who were in isolation or had any experience of isolation to complete the STAI questionnaire. In addition, many workers, business leaders, and students have experienced anxiety due to work, production, and study shutdown during COVID-19. Thus, this study also invited such people to complete the STAI questionnaire through the recommendation of psychological counselors. Before conducting the questionnaire and experiment, each testee was required to sign an informed consent form. The contents of the informed consent form include specific practical steps, research purposes, and time spent. Finally, a total of 63 workers, students, and business managers were invited to fill out the STAI questionnaire; 28 males and 35 females, ranging in age from 23 to 56. This study used Cronbach coefficient α to verify the reliability of the questionnaire results, as shown in Table 3. The Cronbach coefficient α was 0.963, which confirmed the credibility of the questionnaire results. Finally, we selected 20 people with scores ranging from 106 to 137 as testees for this study, 5 males and 15 females. For the follow-up study, we divided the testees into two groups on average, and the two groups evaluated the schemes according to the evaluation criteria and subjective feelings, respectively.

#### 4.1.2. Construct an Information Table of Healing Commodities

To collect more valid research samples, we consulted psychological counselors and conducted extensive internet research. Finally, 100 healing commodities were selected as research samples, as shown in Figure 2. Subsequently, we constructed an information table to understand and distinguish these samples, as shown in Table 4. The information table includes images, features, and materials of healing commodities.

#### 4.1.3. Cluster Analysis of Healing Products

This section aims to perform cluster analysis on 100 healing commodities using FCM. To collect sufficient information about healing commodities, this study proposed seven essential characteristics suitable for describing the healing contribution of samples based on the human five senses, as shown in Table 5.

Subsequently, we scored 100 healing commodities using the proposed essential characteristics (see Table 5) on a scale of 0 to 1. Specifically, 20 testees were invited to read the basic information about all healing commodities (see Table 4). In addition, to ensure that the testees have a more comprehensive understanding of each sample, we presented the sample pictures from different angles and played the operation video of the sample and the sound made by the sample. Then, all testees were invited to score the healing products according to the seven essential characteristics. Finally, we obtained a rating matrix.

Based on the obtained scoring matrix, we used FCM to classify the 100 healing commodities into four clusters. The relevant parameters in the clustering process are: iterations = 1000, threshold = 0.00001, fuzzy parameter *m* = 1.5, and cluster validity = 0.856. In this paper, we used the three samples closest to the center point of each cluster as the representative samples of this cluster. As shown in Figure 3, the stars represent the center point of each cluster and the three numbers around the center point represent representative samples of this cluster. The information about 12 representative samples is shown in Table 6. The characteristic of Cluster_0 (i.e., red dots) is that all samples could relieve users’ anxiety through particular sounds and interactive methods. The characteristic of Cluster_1 (i.e., green dots) is that users could release their anxiety by squeezing samples to release stress. The characteristic of Cluster_2 (i.e., blue dots) is that all samples could create a unique sense of order to relieve users’ anxiety. The characteristic of Cluster_3 (i.e., purple dots) is that all samples could relieve users’ anxiety through color matching and unique interactive methods.

Furthermore, to deeply analyze the attractive factors of healing commodities in the follow-up research we purchased all representative samples from the online sales platform.

### 4.2. Phase 2: Analysis

#### 4.2.1. Build a Three-Layer Diagram for Healing Products

As described in Section 4.1.1, 20 people with moderate or high anxiety were included as testees in this study. This section aims to get a three-layer diagram of healing products through EGM. Specifically, half of the testees were invited to experience these representative samples within a week. Then, we conducted in-depth interviews with each testee, employing on-site interviews. Finally, sorting out the responses of all testees resulted in a three-layer diagram, as shown in Figure 4. The upper layer consists of 8 abstract evaluation items (AEIs, from U1 to U8), the middle layer consists of 11 original evaluation items (OEIs, from M1 to M11), and the lower layer consists of 32 concrete evaluation items (CEIs, from L1 to L32).

#### 4.2.2. Establish Evaluation Criteria

This section aims to screen the evaluation criteria of healing products from the three-layer diagram. Generally, to accurately evaluate a product, comprehensive and specific items should be selected as evaluation criteria. In Figure 4, the AEIs are abstract and the CEIs are specific but numerous. Compared with the AEIs and CEIs, the abstraction degree of OEIs is between both fronts. Additionally, the number of OEI is moderate. Therefore, this study used 11 items (i.e., from M1 to M11) in the middle layer as the evaluation criteria for healing products.

Furthermore, consider that the importance of each evaluation criterion is different; thus, we used AHP to obtain the relative weight of each evaluation criterion. We invited testees to rate the evaluation criteria according to the implementation steps described in Section 2.5. Subsequently, a paired comparison matrix was established with the average score, as shown in Table 7. Finally, the geometric mean method was used to obtain the relative weights of 11 evaluation criteria, as shown on the right side of Table 7. The result of the consistency test is C.R.=C.I./R.I.=0.147/1.51=0.097<0.1. Thus, this paired comparison matrix is acceptable.

#### 4.2.3. Build Design Guidelines

In design practice, design guidelines are crucial reference factors and design basis. In the three-layer diagram, because CEIs are concrete attractiveness factors, this section aims to screen out relatively important items from CEIs as the design guidelines for healing products. First, set up a questionnaire based on the three-layer diagram. Specifically, we set OEIs as questions and CEIs as options. Then, all anxious testees were invited to complete this questionnaire. Additionally, to get the relatively important items in CEIs, we used QTTI to analyze the questionnaire data. Finally, we removed insignificant items and the simplified results are shown in Table 8. The coefficients of determination (R^2^) are all greater than 0.55. Thus, the questionnaire results are acceptable. In this paper, we use 18 items with higher category scores as design guidelines for healing products, namely L1, L2, L3, L6, L7, L8, L9, L10, L12, L13, L14, L15, L16, L18, L21, L28, L30, and L31. If these design guidelines are divided according to the five senses, L7, L8, L9, L10, L12, L14, L21, and L28 belong to sight; L3, L7, and L10 belong to hearing; L1, L2, L3, L6, L12, L13, L14, L15, L16, and L18 belong to touch; L30 and L31 belong to smell. It can be seen that the design guidelines for healing products obtained through EGM, questionnaires, and QTTI are more about touch, sight, and hearing but lack smell and taste.

### 4.3. Phase 3: Verification

#### 4.3.1. Innovative Design

To further verify the effectiveness of the design guidelines and the evaluation criteria, we invited two teams to design four alternatives in this section. Specifically, ten designers were selected to form two teams by evaluating the designers’ comprehensive abilities. The comprehensive ability of the two teams is equal and each team consists of five people. The design task was to ask each team to design two alternatives within a week. The first team was required to refer to the design guidelines (see Table 8). The second team was not given any additional instructions. Finally, the names, images, features, materials, and embodied design guidelines of the four alternatives are shown in Table 9. Among them, A1 and A2 are the design schemes from the first team and A3 and A4 are the design schemes from the second team. The first team referenced the design guidelines, A1 and A2 embody 12 and 8 items from the design guidelines, respectively. In addition, since the second team was not given any instructions, A3 and A4 embody six and seven items from the design guidelines, respectively.

#### 4.3.2. Design Evaluation Based on Evaluation criteria

This section aims to use the GRA to prioritize all schemes in the innovation and reference groups based on the proposed evaluation criteria, thereby confirming the validity and usability of the design guidelines. Precisely, the innovation group consists of four newly designed schemes (see Table 9). As described in Section 4.1.3, all samples are divided into four clusters and each cluster has unique characteristics. Therefore, we constructed a reference group using the four samples closest to each cluster center point, namely P19, P43, P13, and P2 (see Table 6). In this paper, we invited 10 testees to score the eight schemes from two groups using 11 evaluation criteria (see Table 7) and then used the average score of 10 testees as the final grading results. The rating scale is from 0 to 10. The testees were provided with detailed information on eight schemes during the evaluation process, including name, image, feature, and material. Finally, Equation (8) is used to obtain the ideal optimal target sequence P0 (i.e., the reference sequence) based on the scoring results. Eight comparative sequences (i.e., A1, A2, A3, A4, P19, P43, P13, P2) and reference sequence P0 are shown in Table 10.

Furthermore, as described in Section 4.2.2, we have used AHP to obtain a set of relative weights for evaluation criteria (see Table 7), namely *A* = [0.217 0.079 0.231 0.060 0.073 0.059 0.052 0.101 0.067 0.038 0.022]. Thus, to improve the accuracy of GRA, we updated the values of comparative sequences and the reference sequence based on the weights of evaluation criteria and the updated data are shown in Table 11. Subsequently, according to the calculation steps described in Section 2.6, the grey correlation values $r_{i}$ of each comparison sequence and the reference sequence are shown in Table 11. Thus, the priority of the eight schemes is (A1, P43, P19, A2, A4, P13, A3, P2).

#### 4.3.3. Perceptual Questionnaire Evaluation

In Section 4.3.2, we invited ten testees to evaluate eight healing products using 11 evaluation criteria and then obtained their priority. In this section, to confirm the validity of the evaluation results based on the evaluation criteria, we invited the other half of the ten testees to rate the eight healing products using a perceptual evaluation questionnaire. Testees rated the healing power of the eight products based on their subjective feelings on a scale from 0 to 10. Compared with the evaluation based on evaluation criteria, in the process of the perceptual questionnaire evaluation we not only provided the testees with detailed information on eight products but also provided the physical objects of four healing commodities and the dynamic videos of four conceptual design schemes. The final evaluation results are shown in Table 12. The priority of the eight products is (A1, P43, P13, A2, P19, A4, A3, P2).

## 5. Results and Discussions

In the preparation phase, based on the human five senses, we proposed seven essential characteristics suitable for describing the healing contribution of the samples, as shown in Table 5. Although these essential characteristics relate to the color, shape, sound, material, interaction, smell, and taste of the healing product, they are not specific enough. In this paper, they are used for clustering samples, but are not suitable for evaluating samples. Therefore, we obtained 11 more specific evaluation criteria using EGM during the analysis phase.

In the analysis phase, we obtained 18 reference guidelines (i.e., L1, L2, L3, L6, L7, L8, L9, L10, L12, L13, L14, L15, L16, L18, L21, L28, L30, L31) using EGM and QTTI, as shown in Figure 4 and Table 8. As described in Section 4.2.3, although those 18 items dealt with tactile, auditory, and visual, almost none dealt with smell and taste. However, smell and taste are also important senses for human beings to experience the world. Therefore, in the follow-up research we could expand the scope of the sample collection to collect some healing smells and tastes and then provide more comprehensive design guidelines.

In the verification phase, as shown in Table 11 and Table 12, the priority obtained based on the evaluation criteria is (A1, P43, P19, A2, A4, P13, A3, P2) and the priority obtained based on the perceptual questionnaire is (A1, P43, P13, A2, P19, A4, A3, P2). Although there are slight differences between the two evaluation results, there is overall consistency; both A1 and P43 are the top two and A3 and P2 are both the bottom two. Thus, the effectiveness and practicability of the evaluation criteria are confirmed. In other words, healing products could be accurately and quickly evaluated using 11 evaluation criteria. In addition, in the two evaluation results the priority of the four alternatives is (A1, A2, A4, A3). Among them, A1 and A2 are the design schemes of the first team and A3 and A4 are the design schemes of the second team. As described in Section 4.3.1, the first team was required to refer to the design guidelines and the second team was not given any additional instructions. Thus, the effectiveness and practicability of the design guidelines are confirmed; that is, more healing products could be designed by referring to the design guidelines in the design activities. Furthermore, A2 is ranked fourth among all schemes in both evaluation results. The reason may be that A2 does not refer to more design guidelines or the healing products ranked ahead of A2 are already outstanding. Therefore, designers should more actively refer to design guidelines during the development of healing products.

## 6. Conclusions

In the context of the COVID-19 pandemic, we integrated multiple effective research methods to propose a healing product design strategy for anxious users. This design strategy includes three phases, namely the preparation (Phase I), the analysis (Phase II), and the verification (Phase III). In Phase I, to obtain a representative set of healing products, we invited 20 anxiety users to rate 100 healing products. Finally, 12 representative samples were obtained using Fuzzy C-means. In Phase II, to construct evaluation criteria and reference guidelines we obtained a three-layer diagram of healing products using EGM and QTTI. Finally, the evaluation criteria and reference guidelines were constructed with middle- and lower-layer items. In Phase III, to verify the effectiveness and practicability of the evaluation criteria and reference guidelines we invited two teams to design four alternatives and then formed an innovation group. The first team had to actively refer to the design guidelines, while the second team was not given any instructions. Meanwhile, we selected four samples closest to the cluster center to form a reference group. Lastly, eight schemes in the innovation and reference groups were evaluated using two methods. The results showed consistency between the two evaluations, and the two alternatives obtained with reference to the design guidelines were better than the other two alternatives.

The design strategy proposed in this study could be used to develop and evaluate healing products. Overall, we summarize the key contributions of this paper as follows. Other scholars mainly focus on particular groups such as children, older adults, or single women. In contrast, the anxious users in this study were drawn from various groups affected by COVID-19 and were not limited to particular groups. Moreover, the evaluation criteria and design guidelines presented in this study enhance the practicability of this design strategy. Specifically, designers could design better schemes by referring to design guidelines that have been confirmed in this paper. Based on evaluation criteria, psychological counselors could analyze existing healing products more simply and recommend healing products for different anxiety users. Therefore, the proposed design strategy is beneficial to alleviating people’s anxiety during the COVID-19 pandemic.

The present study is subject to some limitations. First, this study collected 100 healing commodities as research samples through internet surveys and expert interviews, but these commodities only involve tactile, auditory, and visual. However, we discovered during the interviews that smell and taste are essential senses. Therefore, we could supplement some perfumes and sweets as samples in future research to obtain more comprehensive and accurate design guidelines. Second, due to time constraints testees only spent a week experiencing 12 representative samples. Thus, if the duration of the experience is extended, we may achieve some more profound descriptions of healing products. Lastly, although we invited two design teams to produce four alternatives and provided various information about them in the evaluation, these four schemes are still conceptual designs. Thus, if the conceptual design could be transformed into a physical product through the 3D printing technology or CNC machine, we may get more accurate research results. Still, much research is needed to improve the design strategy.

## Figures and Tables

**Figure 1 ijerph-19-06046-f001:**
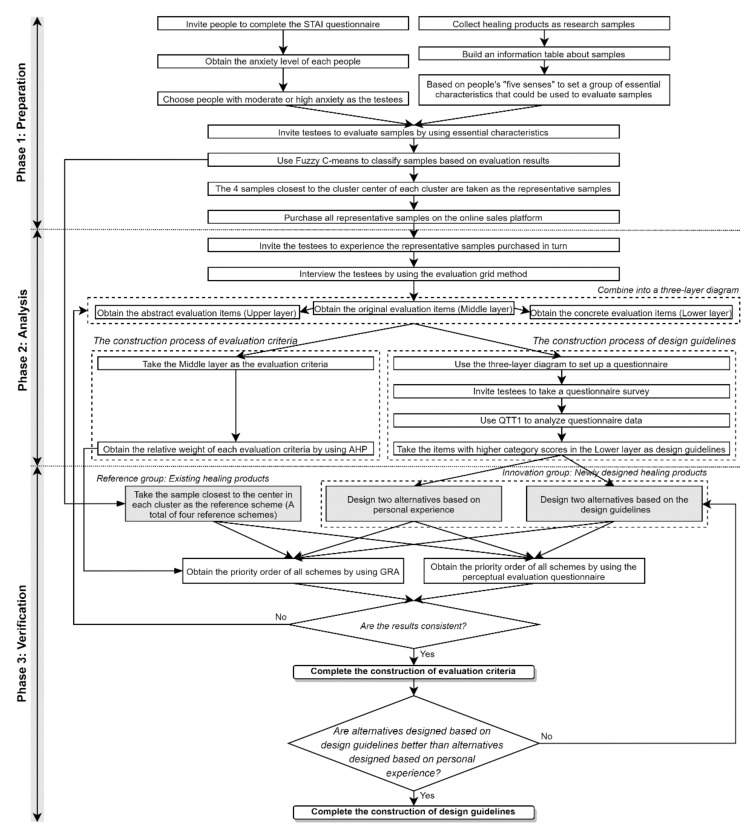
The research framework.

**Figure 2 ijerph-19-06046-f002:**
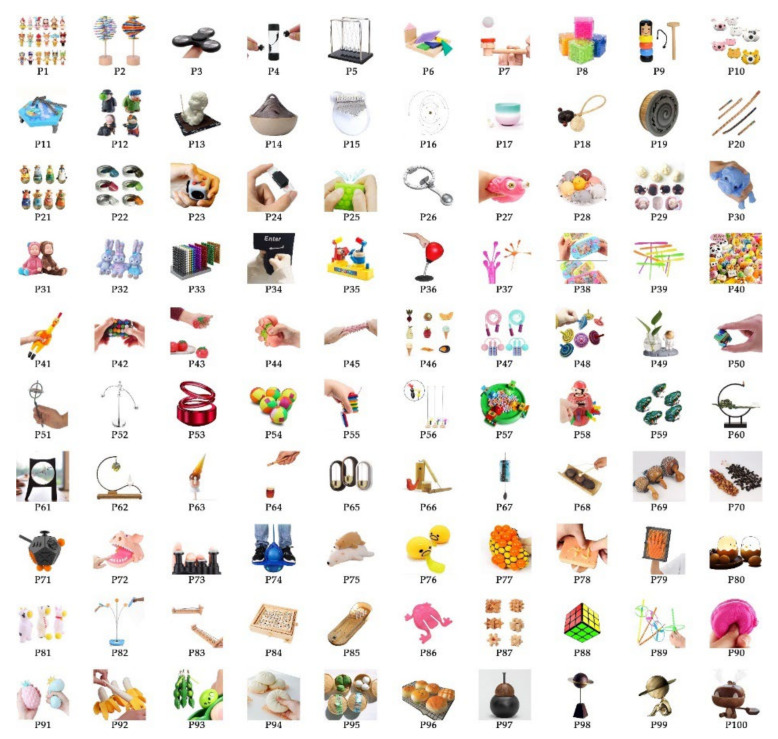
100 images of healing commodities (P1–P100).

**Figure 3 ijerph-19-06046-f003:**
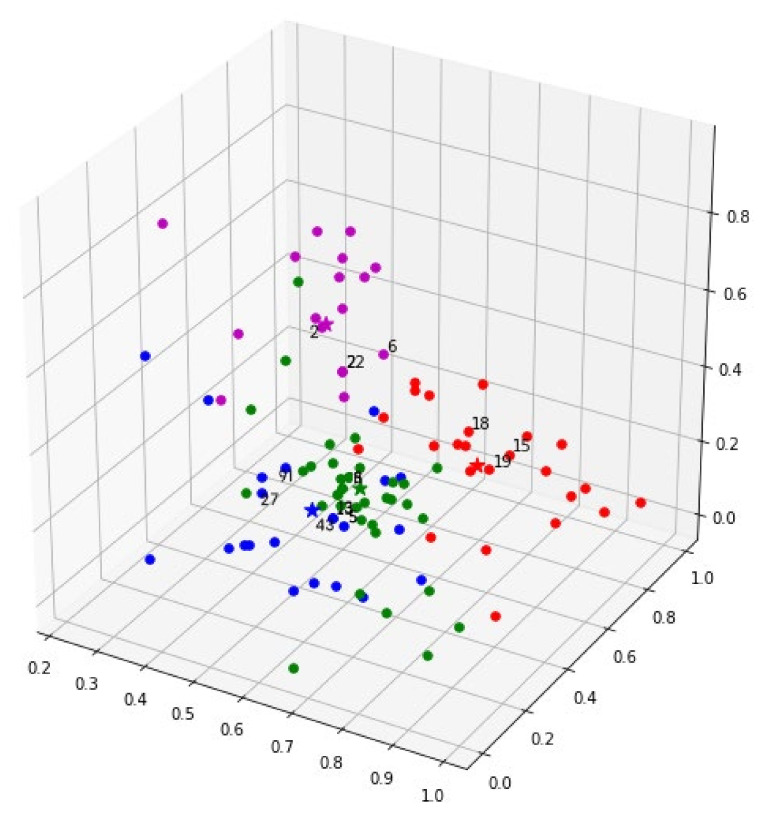
Results of cluster analysis of 100 healing commodities.

**Figure 4 ijerph-19-06046-f004:**
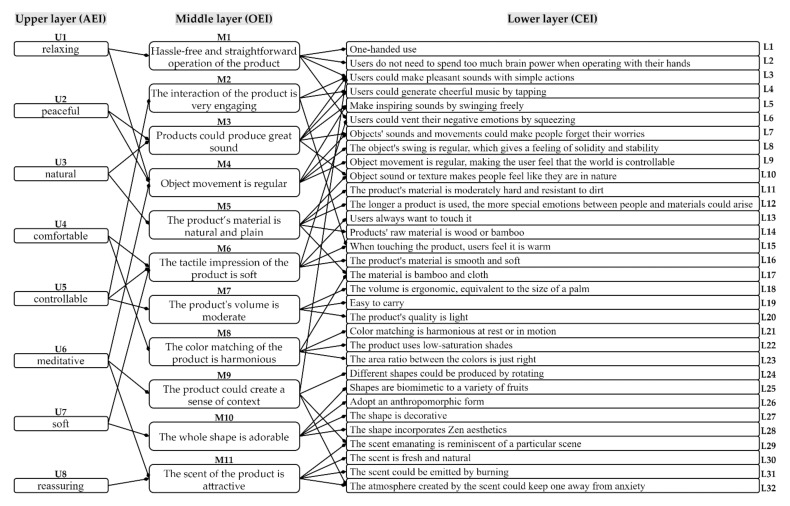
The three-layer diagram of healing products.

**Table 3 ijerph-19-06046-t003:** Cronbach reliability analysis.

Number of Items	Sample Size	Cronbach Coefficient α
40	63	0.963

**Table 4 ijerph-19-06046-t004:** Information table about healing commodities.

Number	Name	Image	Feature	Material
1	Bubble Mart Blind Box	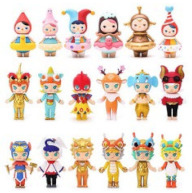	It has a variety of anthropomorphic shapes with different themes, cute shapes, and gorgeous colors. They can be displayed at home for admiration or you can play with them.	resins and plastics
2	Spinning Wooden Lollipops	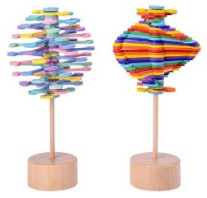	Its shape could be changed from a lollipop to a sphere. The colors are gorgeous and presented in a layered manner. Users could control the change of the lollipop shape by rubbing the wooden stick with their hands and it will make a crisp sound.	wood
⋮	⋮	⋮	⋮	⋮
100	Big mouth dragon aromatherapy stove	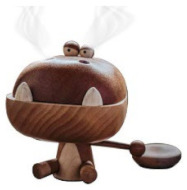	The shape is cute and the color is natural and plain. Incense can be burned inside the big mouth and smoke can come out of the nose.	wood

**Table 5 ijerph-19-06046-t005:** Seven essential characteristics for describing the healing contribution of samples.

Five Senses	Essential Characteristics
Sight	How healing is the color matching?
	How healing is the shape?
Hearing	How healing is the sound made?
Touch	How healing is the texture of the material?
	How healing is the way of interaction?
Smell (metaphorical)	How healing is the scent conveyed?
Taste (metaphorical)	How healing is the flavor conveyed?

**Table 6 ijerph-19-06046-t006:** Information table about representative samples from four clusters.

Cluster	Number and Name	Image	Feature	Material
Cluster_0	P19: Stream Drum	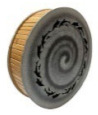	It is made of bamboo and wood and has many seeds of plants inside. Its shape is a cylinder and its color is natural and plain. Users could make a sound similar to running water by shaking it.	wood and bamboo
	P15: Finger Piano	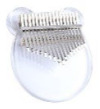	Its shape is small and delicate and the color matching is simple. Users could use the finger piano with one hand and make different tones of sound by dialing the metal bars of different lengths.	metal and acrylic
	P18: Handmade Temple Block	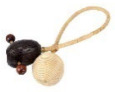	Its raw materials are taken from the fruit of plants, so the shape and color are natural and plain. Users could make it sound by shaking it.	rattan and plant fruit
Cluster_1	P43: Vent Tomatoes	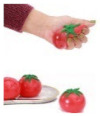	It mimics the shape and color of tomatoes. It has a soft texture and good elasticity. Users could squeeze it with confidence.	elastoplastic and water
	P27: Squeeze Eyes Toy	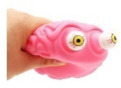	They generally have one or two eyes and have bionic shapes and rich colors. Users could make their eyes stick out by squeezing.	rubber and plastic
	P91: Pinchable Fruit	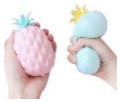	They imitate the shapes of various fruits, with soft textures and simple color matchings. Users could squeeze them at will with their hands.	rubber and pearl blister
Cluster_2	P13: Peace Buddha Aromatherapy	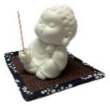	It has a variety of anthropomorphic shapes, such as sitting and lying positions. The shape is round and cute and the color is mainly pure white. Users insert the lit incense into the hand of the character, which can create a Zen atmosphere. The smoke spirals upward to create a sense of order.	ceramics
	P8: Hexahedron Maze	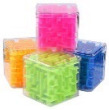	It is a regular hexahedron with a small ball inside it. The six faces are an independent maze and are connected. Users need to mobilize their spatial thinking ability and sense of balance to move the ball out of the maze.	plastic and acrylic
	P5: Oscillating Perpetual Motion	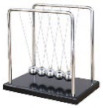	It consists of a base, bracket, metal ball, and connecting wire. Shape and color tend towards industrial style. Users could realize the cyclic swing of the metal balls on both sides by moving the metal balls on either side.	metal, nylon, and acrylic
Cluster_3	P2: Spinning wooden lollipops	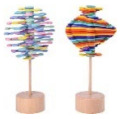	Its shape could be changed from a lollipop to a sphere. The colors are gorgeous and presented in a layered manner. Users could control the change of the lollipop shape by rubbing the wooden stick with their hands and it will make a crisp sound.	wood
	P6: Colorful Tangram	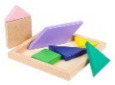	Before splitting, it is a square, and after splitting, it includes multiple triangles, squares, and parallelograms. It consists of multiple colors. By careful observation, users need to put the disassembled blocks of various shapes into the square box.	wood
	P22: Magnetic Plasticine	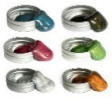	It is a combination of plasticine and magnetic powder. Its shape is changeable and its color is single. Users could move the magnetic plasticine through an iron block.	magnetic powder and plasticine

**Table 7 ijerph-19-06046-t007:** The paired comparison matrix of the evaluation criteria.

	M1	M2	M3	M4	M5	M6	M7	M8	M9	M0	M1	Geometric Mean	Weight
M1	1.00	5.67	2.78	4.33	2.78	3.67	4.11	0.86	3.67	3.67	5.00	2.980	0.217
M2	0.18	1.00	0.24	1.16	2.51	2.11	1.18	0.86	1.22	2.11	3.67	1.088	0.079
M3	0.36	4.17	1.00	4.11	3.44	5.67	4.78	5.67	2.33	5.00	8.33	3.162	0.231
M4	0.23	0.86	0.24	1.00	1.18	2.11	1.18	0.36	1.78	0.48	2.78	0.825	0.060
M5	0.36	0.40	0.29	0.85	1.00	0.29	4.33	0.82	2.11	3.67	3.67	1.002	0.073
M6	0.27	0.47	0.18	0.47	3.44	1.00	1.89	0.47	0.56	2.78	2.07	0.814	0.059
M7	0.24	0.85	0.21	0.85	0.23	0.53	1.00	0.86	1.22	2.11	2.70	0.719	0.052
M8	1.16	1.16	0.18	2.78	1.22	2.11	1.16	1.00	1.18	3.44	4.33	1.382	0.101
M9	0.27	0.82	0.43	0.56	0.47	1.79	0.82	0.85	1.00	3.00	4.33	0.921	0.067
M0	0.27	0.47	0.20	2.07	0.27	0.36	0.47	0.29	0.33	1.00	3.00	0.518	0.038
M11	0.20	0.27	0.12	0.36	0.27	0.48	0.37	0.23	0.23	0.33	1.00	0.303	0.022

**Table 8 ijerph-19-06046-t008:** The analysis results from QTTI.

AEI	OEI	CEI	Category Score	Coefficient of Determination
U1	M1	L1	0.888	R^2^ = 0.794
		L2	0.699	
	M4	L8	0.718	
U2	M3	L10	0.527	R^2^ = 0.803
	M4	L9	0.922	
U3	M3	L3	0.389	R^2^ = 0.616
	M5	L12	0.793	
		L14	0.310	
U4	M6	L16	0.860	R^2^ = 0.714
	M8	L21	0.711	
U5	M2	L15	0.500	R^2^ = 0.55
	M6	L13	0.434	
	M7	L18	0.381	
U6	M4	L7	0.275	R^2^ = 0.723
	M9	L3	0.407	
	M11	L30	0.242	
U7	M10	L28	0.592	R^2^ = 0.55
	M6	L6	0.579	
U8	M11	L31	0.527	R^2^ = 0.62

**Table 9 ijerph-19-06046-t009:** Information table about the four alternatives.

Name	Image	Feature	Material	Design Guidelines
A1: Landscape wind chimes	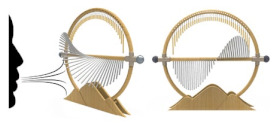	Its shape incorporates Zen elements such as mountains, water, and the sun. Its color is natural and plain, using the inherent color of the material. There are dozens of metal rods of various lengths. Users could make a pleasing sound by blowing or pulling it.	wood and metal	L1, L2, L3, L7, L8, L9, L10, L12, L13, L14, L21, L28 (the amount of 12)
A2: Mood shredder	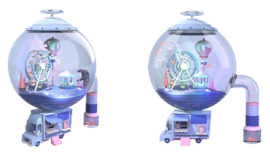	Its shape consists of two parts: the bottom is similar to a vending car and the top is a sphere with a miniature playground inside. Its color is dominated by less saturated purple, supplemented by pink. When users put a piece of paper into the vending cart, it would rustle. Then, the shredded paper is blown into the sphere, creating a scene of snow falling.	plastic, metal, and rubber	L1, L2, L3, L7, L9, L10, L12, L13, L21 (the amount of nine)
A3: Clown’s nose	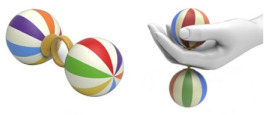	Its shape consists of two spheres and a ring. The two spheres are colorful and resemble a clown’s nose. Both spheres are made of soft material and users can pinch them.	rubber and plastic	L1, L2, L6, L13, L18, L21 (the amount of six)
A4: Interactive squeeze ball	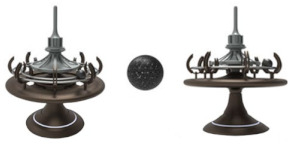	Its shape is similar to an ancient seismograph or incense burner and consists of a tray, a base, a track, a top, and two balls. Its colors are classic. Users could make the other ball rotate on the track by squeezing one ball. The greater the frequency and force of the squeezing the faster the ball spins on the track. Additionally, there is a spinning top at the top of the product.	rubber and plastic	L1, L2, L6, L8, L9, L13, L28 (the amount of seven)

**Table 10 ijerph-19-06046-t010:** Information table about the eight comparative sequences and the reference sequence.

Evaluation Criteria	P0	A1	A2	A3	A4	P19	P43	P13	P2
M1	9.13	8.25	7.88	8.75	6.50	9.13	8.00	8.00	9.13
M2	9.38	9.00	9.38	6.75	8.63	6.38	7.00	5.88	4.13
M3	9.13	9.13	7.38	0.63	3.63	8.88	6.75	0.50	1.13
M4	7.50	7.50	4.63	3.00	6.63	6.88	7.38	3.75	0.88
M5	7.88	7.50	2.75	1.50	4.50	7.88	6.25	6.63	2.50
M6	9.38	0.88	0.88	8.38	4.25	0.88	0.75	1.50	9.38
M7	8.38	6.13	6.00	7.00	6.13	7.63	7.50	7.88	8.38
M8	9.63	7.00	9.63	6.50	4.50	6.38	8.75	7.13	5.75
M9	9.25	8.50	9.25	4.38	6.75	8.00	5.25	8.88	3.50
M10	9.00	5.75	9.00	7.50	4.88	5.75	6.50	8.38	7.50
M11	8.75	0.00	0.00	0.00	0.00	0.25	0.00	8.75	3.38

**Table 11 ijerph-19-06046-t011:** Information table about the eight comparative sequences and the reference sequence after updating.

Evaluation Criteria	P0	A1	A2	A3	A4	P19	P43	P13	P2
M1	1.98	1.79	1.71	1.90	1.41	1.98	1.74	1.74	1.98
M2	0.74	0.71	0.74	0.53	0.68	0.50	0.55	0.46	0.33
M3	2.11	2.11	1.70	0.14	0.84	2.05	1.56	0.12	0.26
M4	0.45	0.45	0.28	0.18	0.40	0.41	0.44	0.23	0.05
M5	0.57	0.55	0.20	0.11	0.33	0.57	0.46	0.48	0.18
M6	0.55	0.05	0.05	0.49	0.25	0.05	0.04	0.09	0.55
M7	0.44	0.32	0.31	0.36	0.32	0.40	0.39	0.41	0.44
M8	0.97	0.71	0.97	0.66	0.45	0.64	0.88	0.72	0.58
M9	0.62	0.57	0.62	0.29	0.45	0.54	0.35	0.59	0.23
M10	0.34	0.22	0.34	0.29	0.19	0.22	0.25	0.32	0.29
M11	0.19	0.00	0.00	0.00	0.00	0.01	0.00	0.19	0.07
Grey relation ($r_{i}$)		0.896	0.869	0.808	0.860	0.886	0.889	0.855	0.796
Ranking		1	4	7	5	3	2	6	8

**Table 12 ijerph-19-06046-t012:** The result of perceptual evaluation for the eight healing products.

	A1	A2	A3	A4	P19	P43	P13	P2
Total score	86	77	72	73	75	83	80	69
Ranking	1	4	7	6	5	2	3	8

## Data Availability

Not applicable.

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
