# Peer review of "Research on the Design Strategy of Healing Products for Anxious Users during COVID-19"

_ijerph, 2022, doi:10.3390/ijerph19106046_

Round 1
Reviewer 1 Report
Thank you for the opportunity to review the article” Research on the Design Strategy of Healing Products for Anxious users During the Covid-19”
Overall, it is a good manuscript, however there are a few specific issues, to be addressed as follows:
1) Abstract:
- Describe the sample
2) Introduction
- It is a good introduction. The introduction briefly describes the study and purpose. It also includes the current state of the research field providing sufficient background, and the main aim of the work.
3) Methods
- Please describe the sample age
- Please describe the informed consent
4) Results
- In figure 3 describe what color corresponds to each cluster
Author Response
|
|
Comments from reviewer 1 |
Response |
|
1 |
Abstract: Describe the sample |
Thanks for your comments. We have briefly addressed testees and samples in the abstract of the revised manuscript. Please see lines 13 and 14 in the revised manuscript. Subsequently, we described the source of the testees in detail in section 4.1.1. Please see lines 299 to 302 in the revised manuscript. In section 4.1.2, we described the source of the samples in detail and constructed a sample information table (i.e., Table 4). Please see lines 317 to 319 in the revised manuscript. |
|
2 |
Introduction: It is a good introduction. The introduction briefly describes the study and purpose. It also includes the current state of the research field providing sufficient background, and the main aim of the work. |
Thanks for your encouragement. As you mentioned, the 5 paragraphs in the introduction are: Paragraph 1: Introduction to anxiety caused by COVID-19; Paragraph 2: Starting from the five senses can reduce anxiety; Paragraph 3: Healing products based on the five senses can reduce anxiety; Paragraph 4: Few scholars have proposed a practical healing product design strategy; Paragraph 5: Introduce the purpose of this study and the organization of the following. |
|
3 |
Methods: Please describe the sample age. Please describe the informed consent.
|
Thanks for your comments. We have supplemented a description of the age of the testees and the informed consent form in the revised manuscript. Please see lines 304 to 308 in the revised manuscript. In addition, we have also provided detailed informed consent to the editors of this journal. |
|
4 |
Results: In figure 3 describe what color corresponds to each cluster. |
Thanks for your comments. We have added a description of the relationship between clusters and colors in the revised manuscript. Please see lines 345 to 350 in the revised manuscript. |
Reviewer 2 Report
The authors offered a design strategy for healing products, which 12 includes three phases: preparation, analysis, and verification. The paper is innovative and interesting but too data have been showed and the study design is difficult to follow, even if the flowcharts are useful. The general aims were not explained in depth, but only a speed mention. They should be more developed.
I found difficult to follow the theoretical background because there are examples of instruments, statistical design/method, procedure explanation, all together. I think that this type of presentation is confounding, even if I think that adopting a more clear explanation of aims could help.
How many participants were involved at the beginning? What about the drop out? What about the procedure to invite people? Describe it. How did you administer the healing commodities? By video or in presence? It isn’t clear.
Which are the levels of anxiety at the starting project and then after the design guidelines? It is not clear how the intervention had its effectiveness.
Discussion and conclusion were limited, and they didn’t clarify the impact of the adopting of these healing products to dampen the anxiety levels.
Round 2
Reviewer 2 Report
I found the paper really ameliorated and more clear. The authors responded to all my comments.
I think that the paper could be publishable.